# A Second Trimester Prediction Algorithm for Early-Onset Hypertensive Disorders of Pregnancy Occurrence and Severity Based on Soluble fms-like Tyrosine Kinase 1 (sFlt-1)/Placental Growth Factor (PlGF) Ratio and Uterine Doppler Ultrasound in Women at Risk

**DOI:** 10.3390/children11040468

**Published:** 2024-04-14

**Authors:** Cristian Nicolae Chirilă, Claudiu Mărginean, Dana Valentina Ghiga, Septimiu Voidăzan, Paula Maria Chirilă, Mirela Liana Gliga

**Affiliations:** 1Department of Internal Medicine-Nephrology, George Emil Palade University of Medicine, Pharmacy, Science and Technology of Târgu Mureș, 540142 Târgu Mureș, Romania; cristian.chirila@umfst.ro (C.N.C.); mirela.gliga@umfst.ro (M.L.G.); 2Department of Nephrology, Mures Clinical County Hospital, 540103 Târgu Mureș, Romania; 3Doctoral School, George Emil Palade University of Medicine, Pharmacy, Science and Technology of Târgu Mureș, 540142 Târgu Mureș, Romania; paulachirila@yahoo.com; 4Department of Obstetrics and Gynecology 2, George Emil Palade University of Medicine, Pharmacy, Science and Technology of Târgu Mureș, 540142 Târgu Mureș, Romania; 5Department of Obstetrics and Gynecology, Mures Clinical County Hospital, 540057 Târgu Mureș, Romania; 6Department of Scientific Medical Research Methodology, George Emil Palade University of Medicine, Pharmacy, Science and Technology of Târgu Mureș, 540142 Târgu Mureș, Romania; dana.ghiga@umfst.ro; 7Department of Epidemiology, George Emil Palade University of Medicine, Pharmacy, Science and Technology of Târgu Mureș, 540142 Târgu Mureș, Romania; septimiu.voidazan@umfst.ro; 8Department of Endocrinology, Mures Clinical County Hospital, 540142 Târgu Mureș, Romania

**Keywords:** preeclampsia, gestational hypertension, soluble fms-like tyrosine kinase 1 (sFlt-1)/placental growth factor (PlGF) ratio, uterine Doppler, intrauterine growth restriction, prediction

## Abstract

Hypertensive disorders of pregnancy (HDPs) represent a significant source of severe maternal and fetal morbidity. Screening strategies relying on traditional medical history and clinical risk factors have traditionally shown relatively modest performance, mainly in the prediction of preeclampsia, displaying a sensitivity of 37% for the early-onset form and 29% for the late-onset form. The development of more accurate predictive and diagnostic models of preeclampsia in the early stages of pregnancy represents a matter of high priority. The aim of the present paper is to create an effective second trimester prediction algorithm of early-onset HDP occurrence and severity, by combining the following two biochemical markers: a soluble fms-like tyrosine kinase 1 (sFlt-1)/placental growth factor (PlGF) ratio and uterine artery Doppler ultrasound parameters, namely the pulsatility index (PI) and the resistivity index (RI), in a population of high-risk pregnant women, initially assessed through traditional risk factors. A prospective single-center observational longitudinal study was conducted, in which 100 women with singleton pregnancy and traditional clinical and medical history risk factors for preeclampsia were enrolled at 24 weeks of gestation. Shortly after study enrollment, all women had their sFlt-1 and PlGF levels and mean uterine artery PI and RI determined. All pregnancies were followed up until delivery. Receiver operating characteristic (ROC) analysis established algorithms based on cutoffs for the prediction of the later development of preeclampsia: PI 1.25 (96.15% sensitivity, 86.49% specificity), RI 0.62 (84.6% sensitivity, 89.2% specificity) and sFlt-1/PlGF ratio 59.55 (100% sensitivity, 89.2% specificity). The sFlt-1/PlGF ratio was the best predictor for preeclampsia, as it displayed the highest area under the curve (AUC) of 0.973. The prediction algorithm for the severe form of preeclampsia, complicated by fetal growth restriction leading to preterm birth, antepartum fetal demise or acute fetal distress with a cerebro-placental ratio of <one consisted of the following cutoffs: PI 1.44 (93.75% sensitivity, 95.24% specificity), RI 0.69 (87.5% sensitivity, 100% specificity) and sFlt-1/PlGF ratio 102.74 (93.75% sensitivity, 95.2% specificity). These algorithms may significantly enhance the prediction accuracy of preeclampsia compared to traditional risk factors. The combination of the sFlt-1/PlGF ratio with mean uterine PI and RI in particular displayed an improved performance in the prediction of severe preeclampsia with the above-mentioned complications, compared to the biochemical markers or uterine Doppler parameters used alone. Therefore, HDP screening strategies should increasingly focus on implementing such algorithms for women who are initially regarded as high risk based on traditional risk factors, in order to properly diagnose HDP and properly limit or manage the later maternal and fetal complications.

## 1. Introduction

Hypertensive disorders of pregnancy (HDPs) represent the second global cause of maternal mortality, following hemorrhage [1]. As far as preeclampsia is concerned, the World Health Organization estimated its prevalence at 2–10% globally, with a current increasing tendency [2]. Depending on the time of onset during pregnancy, preeclampsia may be categorized as early onset (<34 weeks) and late onset (≥34 weeks) [3]. Furthermore, if maternal blood pressure becomes higher than 160/100 mmHg or multiorgan involvement signs occur, preeclampsia is regarded as severe [4].

While there are both traditional medical history factors (thrombophilia, previous pregnancy complicated with preeclampsia, or a first-degree relative diagnosed with preeclampsia) and traditional clinical factors (multifetal gestation, in vitro fertilization, diabetes mellitus, advanced maternal age of >40 years, obesity, preexisting chronic hypertension or chronic kidney disease, systemic lupus erythematosus, or elevated serum uric acid levels or uric acid to creatinine ratio) known to increase the risk of preeclampsia, it is notable that preeclampsia mainly affects healthy nulliparous women who do not display any of the predisposing factors mentioned above [5,6,7]. These medical history and clinical factors have traditionally shown relatively modest performance in the prediction of preeclampsia, with a sensitivity of 37% for the early-onset form and 29% for the late-onset form of the condition [4].

However, as preeclampsia represents one of the major causes of serious and sometimes life-threatening maternal and fetal complications, developing more accurate prediction models of preeclampsia as early in pregnancy as possible represents a matter of high priority. Even apparently mild forms of preeclampsia can deteriorate rapidly and without warning signs [3,8]. In this regard, plenty of recent studies have focused on the role of both biochemical markers and Doppler velocimetry parameters in the early prediction and diagnosis of preeclampsia.

As far as the biochemical markers are concerned, two serum proteins—soluble fms-like tyrosine kinase 1 (sFlt-1), an antiangiogenic protein, and placental growth factor (PlGF), a proangiogenic protein—have displayed increased performance compared to the traditional medical history and clinical factors in predicting HDPs, mainly preeclampsia and some of its related complications, such as fetal growth restriction (FGR), antepartum fetal demise or preterm delivery [9]. sFlt-1 acts as an angiogenesis inhibitor by binding two promoters of physiological vascular growth and proliferation—the vascular endothelial growth factor (VEGF) and PlGF. sFlt-1 is an important regulatory marker of angiogenesis in various tissues throughout the body [10,11], but its overexpression, in combination with the underexpression of PlGF, typically imbalances normal placental angiogenesis, thus creating the conditions for later occurrences of HDPs during pregnancy, especially de novo preeclampsia [12,13]. Generally, sFlt-1 values in the maternal blood begin to increase 4–5 weeks before the onset of preeclampsia; meanwhile, PlGF serum concentration begins to reduce 9–11 weeks before the clinical onset. It is worth mentioning that sFlt-1/PlGF ratio has shown superior performance in predicting, diagnosing and monitoring established preeclampsia and other placenta-related disorders in pregnancy, compared to the two markers alone. The sFlt-1/PlGF ratio represents a valuable tool in ruling out preeclampsia for up to 4 weeks after the assessment. Furthermore, the sFlt-1/PlGF ratio proved to be a valuable tool in the second trimester or early third trimester for the prediction of late-onset preeclampsia and its related complications [14].

Additionally, recent literature data prove that uterine artery Doppler ultrasound has been considered more and more to be an effective tool for the screening of preeclampsia. Before the onset of preeclampsia, resistance to flow within the uterine arteries increases, which may cause an abnormal waveform aspect and increased resistivity (RI) and pulsatility (PI) indices [4,15]. The 2018 International Society of Ultrasound in Medicine and Biology (ISUOG) Practice Guidelines considered PI to be the gold standard parameter in preeclampsia screening. At 23 weeks of pregnancy, the 95th centile of the uterine artery PI has been established as 1.44 by transabdominal ultrasound. Moreover, among pregnant patients with a mean uterine PI > 90th centile, an increased prevalence of functionally significant cardiac defects, which were not previously diagnosed, could be noticed [16]. The systolic/diastolic ratio (S/D) and RI are also valuable parameters in preeclampsia screening and monitoring. Abdel-Razik provided a RI cutoff of 0.61 at the 20–24 weeks scan for the later development of preeclampsia [17]. The screening of uterine arteries at 20 weeks by Doppler ultrasound had also detected 73.3% of cases with early-onset FGR in a study group of patients without the traditional risk factors for preeclampsia [18].

Ideally, initial screening for preeclampsia should be performed between 11 and 14 weeks. This first trimester screening should assess, first of all, the presence of traditional risk factors [4], as well as mean arterial pressure, uterine artery PI and biochemical markers determined from the maternal blood—PlGF and PAPP-A (pregnancy-associated plasma protein A). This would enable physicians to start early prophylaxis with low-dose aspirin in pregnancies considered to be at high risk or moderate risk [14,19].

The aim of the current study is to assess the utility of a potential second trimester prediction algorithm of early-onset HDP occurrence and severity, with a particular focus on preeclampsia, consisting of a combination of biochemical markers and uterine Doppler ultrasound in a population of high-risk pregnant women, initially assessed by clinical and medical history risk factors.

## 2. Materials and Methods

### 2.1. Study Population

We conducted a prospective single-center observational longitudinal study over a period of 16 months, from January 2021 until April 2022, in the Obstetrics and Gynecology Department of Mures Clinical County Hospital, Targu Mures, Romania. The study sample consisted of 100 women with singleton pregnancy, enrolled at 24 weeks of gestation, initially regarded as carrying a highrisk of later development of preeclampsia from a clinical point of view—based on the presence of at least one of the traditional clinical or medical history risk factors. Consequently, inclusion criteria for patients in the study group were represented by the existence of a previous personal diagnosis of type 2 diabetes mellitus, a previous pregnancy complicated with preeclampsia, a first-degree relative with medical history of preeclampsia or an advanced maternal age of >40 years. Meanwhile, exclusion criteria consisted of any form of HDP present at the moment of study enrollment, a gestational age lower or higher than 24 weeks, multiple pregnancy, a refusal to sign the informed consent or non-compliance with the periodic obstetrical follow-up consultations. It is worth mentioning that the enrolled patients did not benefit from a standard first trimester preeclampsia screening based on traditional risk factors, blood pressure values, biochemical markers or uterine Doppler, nor did they receive prophylaxis with low-dose acetylsalicylic acid. All pregnant women enrolled in the study had their sFlt-1 and PlGF levels determined once, at 24 weeks of gestation from a venous blood sample, when included in the study. For the study purpose, we determined the sFlt-1/PlGF ratio in each case. A Doppler ultrasound evaluation of the uterine arteries was also performed in each case at the moment of inclusion, during the routine ultrasound scan at 24 weeks of gestation, using a transabdominal approach. Furthermore, all patients underwent regular ultrasound Doppler examinations of the umbilical and middle cerebral arteries, including the assessment of the cerebro-placental ratio (CPR), in the moment of enrollment in the study and every four weeks after that, as part of the screening process for FGR and other HDP-related possible complications. All the patients enrolled were followed-up until delivery. A special focus was placed on the cases who developed gestational hypertension or preeclampsia during the later stages of pregnancy, as well as HDP-related complications, such as early-onset FGR and antepartum fetal demise. Fetuses with early FGR matched at least one of the three following criteria for the positive diagnosis, as recommended by the 2020 ISUOG Practice Guidelines [20]: an ultrasound estimated weight < 3rd centile or a uterine artery PI or umbilical artery PI of >95th centile for their gestational age at the moment of the ultrasound scan at a gestational age lower than 32 weeks, with no evidence of congenital anomalies. Pregnant women who did not develop high blood pressure values from the moment of enrollment until at least after birth were considered the control group, whereas patients who developed elevated blood pressure (values ≥ 140/90 mmHg) during the pregnancy, after the inclusion in the study, but with no other criterion for preeclampsia, were considered part of the gestational hypertension group. Women who developed hypertension alongside at least one other diagnostic criterion for preeclampsia, as stated by The American College of Obstetricians and Gynecologists [4] (proteinuria—defined by 24 h urine excretion of a minimum of 300 mg proteins or a spot urinary protein to creatinine ratio of a ≥0.3; thrombocytopenia—expressed by a platelet count of <100,000/microliter; newly developed headache refractory to medication; or a degree of kidney failure with serum creatinine levels of >1.1 mg/dL), during the pregnancy and after study enrollment, were considered part of the preeclampsia group. Lastly, for better quantification of the obstetrical risk in the preeclampsia group, patients who developed a blood pressure of ≥160/110 mmHg, as well as maternal signs such as thrombocytopenia, newly developed headache or kidney failure alongside FGR, leading to preterm birth, antepartum fetal demise or acute fetal distress with a CPR of < one, were categorized as suffering from severe preeclampsia. Meanwhile, the rest of pregnant women from the preeclampsia group, who displayed blood pressure values between 140/90 and 160/110 mmHg and proteinuria, with none of the maternal signs and fetal complications mentioned above, were considered as having mild preeclampsia.

### 2.2. Soluble fms-Like Tyrosine Kinase 1 (sFlt-1) and Placental Growth Factor (PlGF) Assessment

5 mL of a venous blood sample, collected in a biochemistry vacuum blood tube with clot activator and red cap, was obtained from each pregnant women in the Obstetrics and Gynecology Department of Mures Clinical County Hospital, immediately after enrollment in the study. The samples were immediately sent to the attached laboratory and centrifuged in the next 20 min. The serum was stored in 1.5 mL Eppendorf tubes and frozen at −20 °C for a few days. Two serum samples were obtained from each patient. The serum samples were later sent to the Advanced Center of Medical and Pharmaceutical Research of the University of Medicine, Pharmacy, Science and Technology of Targu Mures, Romania, where they were stored in a freezer at −80 °C for a maximum of 9 months. sFlt-1 and PlGF levels were then determined from these samples using a sandwich enzyme-linked immunosorbent assay (ELISA) technique, with the aid of an Elisa Dynex DSX fully automated ELISA analyzer.

### 2.3. Uterine Arteries Doppler Ultrasound

Doppler ultrasound scans of the uterine arteries were performed by a single experienced physician, using a Voluson E8 BT18 ultrasound device (General Electric Healthcare, Chicago, IL, USA) with a RAB-6D convex volumetric abdominal probe.

The second trimester transabdominal ultrasound examination technique was performed according to the protocol issued by ISUOG in 2018 [16], consisting of the following steps: firstly, a clear image of the uterus and cervical canal was obtained in a sagittal section; then, the transducer was placed in the right iliac fossae, tilted towards the right lateral wall of the uterus and towards the pelvis. The central or lateral position of the placenta was noticed. Color flow mode was activated and the probe was moved on the right side, maintaining its medial angulation at all times, until the right uterine artery was identified as it crossed the right external iliac artery. We set a narrow Doppler sampling gate of 2 mm and moved it on the right uterine artery at a 1 cm distance from the point of the apparent intersection with the right external iliac artery. While assuring a high-quality view of the right uterine artery with an insonation angle of <50°, the pulsed wave Doppler mode was activated and at least three identical consecutive waveforms of the right uterine artery were obtained. Finally, Doppler parameters PI and RI were measured. The same steps were then used, in order to measure left uterine artery PI and RI, with the probe placed in the left iliac fossae (Figure 1). Mean uterine artery PI and RI were determined as an arithmetic mean of the left and right values in each case. The mean values were used in order to develop the HDP prediction algorithm.

The normal range of mean uterine artery Doppler PI values was considered to be 1.07 ± 0.38, as per the recommendations provided by Cavoretto [21], while PI values over 1.44 were considered to be a significant risk factor for preeclampsia, as per the 2018 ISUOG Practice Guidelines recommendations [16].

### 2.4. Statistical Analysis

Statistical protocol was realized with the aid of International Busines Machines (IBM) Statistical Package for Social Sciences (SPSS) Statistics 23, MedCalc version 7 software and Microsoft Excel (v14.0). Descriptive statistics were applied for the following three continuous variables: mean uterine PI, mean uterine RI values and sFlt-1/PlGF ratio. The mean and standard deviation (SD) or the median and range were reported for these quantitative variables. The five categorical variables (preeclampsia, mild preeclampsia, severe preeclampsia, gestational hypertension and control groups) were represented as numbers and percentages. Then, a Shapiro–Wilk test was applied, in order to assess the distribution of the continuous variables. The non-parametric Kruskal–Wallis test was later used for the median comparison of the three continuous variables. As the results of the Kruskal–Wallis test were statistically significant (statistical significance was considered if *p* value was <0.05 and the confidence interval (CI) was set at 95%), Dunn’s test was conducted, in order to determine exactly which groups displayed statistically significant differences in the median values. The area under the ROC curve (AUC) and sensitivity were calculated, in order to establish the best cutoff values for each parameter (sFlt-1/PlGF ratio, uterine PI and RI). Specificity, positive likelihood ratio (LR+) and negative likelihood ratio (LR−) were also evaluated for each cutoff. AUC values of the three parameters were compared and the variable with the highest AUC value was established as the best predictive test for both preeclampsia and gestational hypertension. Finally, we developed a prediction algorithm for early-onset preeclampsia, gestational hypertension and HDP-related complications (FGR, antepartum fetal demise, and preterm birth) by combining sFlt-1/PlGF ratio values with mean uterine Doppler ultrasound indices PI and RI.

## 3. Results

Out of the 100 patients with traditional risk factors for preeclampsia, included in the study at 24 weeks of gestation, 40 were later diagnosed with gestational hypertension during pregnancy, while 26 received a diagnosis of preeclampsia and 34 pregnant women remained normotensive at least until after delivery. Among women diagnosed with preeclampsia, 10 were categorized as having the mild form, whereas 16 cases developed the severe form. No evidence of Hemolysis, Elevated Liver Enzyme Levels and Low Platelet Levels (HELLP) syndrome or evolution to eclampsia was found among the study sample. The prevalence of these five above-mentioned groups among the total study sample is represented in Table 1 and Table 2, respectively.

It is worth mentioning that all pregnancies with severe preeclampsia were affected by FGR, with signs of acute fetal distress that led to premature delivery and, in 3 out of 16 cases, antepartum fetal demise.

The most relevant demographic data and clinical characteristics of the studied groups are presented in Table 3. All severe preeclampsia cases delivered through caesarean section, except the three pregnancies complicated by antepartum fetal demise, in whom labor was induced with systemic oxytocin. Active observation with periodic ultrasound checks until term was the method of choice in the cases of the gestational hypertension and mild preeclampsia groups, and no preterm delivery was required, as there were no fetal distress signs. However, caesarean delivery was the method of choice in 80% of mild preeclampsia cases.

As far as the uterine Doppler ultrasound’s ability at 24 weeks to predict later preeclampsia occurrence is concerned, the Kruskal–Wallis test revealed that median PI values were significantly different among the control, hypertension and preeclampsia groups (1.15 vs. 1.18 vs. 1.58, *p* < 0.0001), as depicted in Table 4. Furthermore, the results of Dunn’s test showed that median PI values among patients who later developed preeclampsia were significantly higher than in the control group and the gestational hypertension group, as represented in Table 5.

Moreover, at 24 weeks of gestation, RI should also be regarded as a reliable second trimester ultrasound predictor of later preeclampsia. In the same manner as PI, median RI values were significantly different among the control, hypertension and preeclampsia group (0.52 vs. 0.52 vs. 0.7, *p* < 0.0001), as depicted in Table 6. Moreover, Dunn’s test revealed median RI values to be significantly higher in the preeclampsia group compared to both the control and gestational hypertension groups (Table 7).

The ROC analysis established the value of 1.25 as the best mean uterine PI cutoff at 24 weeks for preeclampsia prediction (Figure 2), as it proved to be the value with the best combination of sensitivity and specificity (96.15% and 86.49%, respectively) and a LR+ of 7.12. In a similar manner, the value of 0.62 proved to be the most accurate mean uterine RI cutoff at 24 weeks for preeclampsia prediction, as depicted in Figure 3.

The sFlt-1/PlGF ratio proved to be the most accurate parameter at 24 weeks for preeclampsia prediction among the three variables. The Kruskal–Wallis test emphasized significantly different median values of the ratio between the control, hypertension and preeclampsia groups (10.05 vs. 36.51 vs. 140.7, *p* < 0.0001), as seen in Table 8. Following the same pattern as the Doppler ultrasound parameters, sFlt-1/PlGF ratio values were significantly higher among the preeclampsia group than among both the control and gestational hypertension groups (Table 9).

The ROC analysis revealed that the sFlt-1/PlGF ratio’s cutoff value of 59.55 was the best second trimester predictor tool for preeclampsia. Figure 4 and Figure 5 show this cutoff to be almost perfect, with a sensitivity of 100%, meaning that none of the 26 preeclampsia cases displayed sFlt-1/PlGF values < 59.55 at the time of screening. Table 10 compares AUC values in the cases of the three parameters, supporting the superiority of the sFlt-1/PlGF ratio for preeclampsia prediction, as it displayed the highest AUC value among the three studied variables.

Moreover, from the physicians’ point of view, in parallel to the accurate anticipation of preeclampsia, special focus should be placed on the early prediction of the severe form of preeclampsia. For this purpose, we implemented new cutoff values for the three parameters.

To begin with, among the preeclampsia group, statistically significant differences could be highlighted, with the aid of the Kruskal–Wallis test and Dunn’s test, between median uterine PI values at 24 weeks in the cases of women who later developed the severe form compared to the mild form (1.6 vs. 1.42, *p* < 0.0001). As expected, women from the severe preeclampsia group had the highest median PI value at the time of screening (Table 11 and Table 12).

The same pattern was followed by uterine RI values at the 24 weeks ultrasound scan: patients from the severe preeclampsia group presented significantly higher values than the rest of the sample (0.71 vs. 0.62 in case of mild preeclampsia, *p* < 0.0003) (Table 13 and Table 14).

The ROC analysis established the most accurate Doppler velocimetry indices cutoff values for the prediction of severe preeclampsia, as presented in Figure 6 and Figure 7. In case of mean uterine PI, 1.44 was the value with the best combination of sensitivity and specificity (93.75 and 95.24, respectively). As far as mean uterine RI was concerned, a cutoff value of 0.69 was established, using the same ROC analysis protocol.

The degree of sFlt-1/PlGF ratio imbalance appeared to have a direct influence on the severity of later preeclampsia. The Kruskal–Wallis test emphasized significantly different median values of the ratio among the four groups of the study sample (10.05 vs. 36.51 vs. 140.7, *p* < 0.0001), as seen in Table 15. Following the same pattern as the Doppler ultrasound parameters, sFlt-1/PlGF ratio values were significantly higher in the severe preeclampsia group, compared to the mild preeclampsia group (179.9 vs. 90.48, *p* < 0.0001) and the other two groups (Table 16).

Additionally, the ROC curve analysis set 102.74 as the most accurate cutoff value, as depicted in Figure 8. It is worth mentioning that the mean uterine PI and sFlt-1/PlGF ratio cutoffs displayed the same sensitivity and specificity and, therefore, had the same prognostic value for severe preeclampsia prediction and the same AUC. However, in our study, out of 16 severe preeclampsia cases, 14 had both the mean uterine PI and sFlt-1/PlGF values > the cutoff value, but, in one case, the sFlt-1/PlGF ratio was higher than the cutoff, while the mean uterine PI value was under the cutoff. On the other hand, another case with the later development of severe preeclampsia displayed a sFlt-1/PlGF ratio under the cutoff value, while the mean uterine PI value was higher at 24 weeks of gestation.

As far as the roles of the three markers in the prediction of gestational hypertension are concerned, the Kruskal–Wallis test revealed significantly lower median uterine PI values at 24 weeks, when compared to the preeclampsia group (1.18 vs. 1.58, *p* < 0.0001), as represented in Table 4. Although mean uterine PI values were higher in case of the later hypertensive women compared to the control group, the difference was not statistically significant (*p* = 0.0504) (Table 5). Regarding the sFlt-1/PlGF ratio’s ability to predict gestational hypertension, Dunn’s test revealed significantly higher median values among women later affected by gestational hypertension (36.51 vs. 10.05, *p* < 0.0001) compared to the control group, as well as significantly lower median values among women later affected by gestational hypertension, compared to the preeclampsia group (36.51 vs. 140.7, *p* < 0.0001).

The ROC analysis set the following predictive cutoff values at 24 weeks for gestational hypertension: 1.44 in the case of mean uterine PI (97.5% sensitivity and 30% specificity), 0.74 in the case of mean uterine RI (70% sensitivity and 53.3% specificity) and 59.55 for the sFlt-1/PlGF ratio (85% sensitivity and 46.7% specificity). A comparative analysis of the AUC values between the three variables highlighted that mean uterine RI was the best predictive test for gestational hypertension, as depicted in Table 17 and Figure 9.

Lastly, the accuracy of preeclampsia occurrence and severity prediction provided by using an algorithm based on the combination of sFlt-1/PlGF ratio and mean uterine PI and RI cutoffs proved to be superior when compared to the accuracy of disease prediction provided by traditional medical history and clinical risk factors (mentioned in the Introduction section): out of 100 pregnant women initially enrolled in the study, judged by traditional risk factors for the disease, only 26 women later developed preeclampsia, corresponding to a sensitivity of 26%. On the other hand, if we evaluated the risk of later occurrence of preeclampsia using the three markers’ cutoff values, all 26 cases presented at 24 weeks at least one value of the three parameters that was higher than the cutoff, corresponding to a sensitivity of 100%. At the same time, a similar pattern could be applied for the prediction of severe preeclampsia: only 16 pregnancies were later affected by severe preeclampsia, meaning a sensitivity of 16% in the case of prediction by traditional risk factors at 24 weeks of gestation; meanwhile, the sensitivity became 100% if the prediction of severe preeclampsia was based on an algorithm comprising the specific cutoffs of the three markers, as long as all 16 patients later diagnosed with the severe form displayed at least one marker higher than the cutoff value at the screening time.

## 4. Discussion

The prediction of HDPs has represented an important research theme in recent years, given the numerous benefits brought by proper risk assessment and timely screening of HDPs: allowing better risk stratification and the possibility of implementing personalized, more intensive monitoring programs for women at high risk, of starting early prophylaxis of preeclampsia with low-dose acetylsalicylic acid, of initiating antihypertensive treatment as early as possible once the positive diagnosis is established, in order to avoid or significantly reduce the possibility of complications such as an evolution to eclampsia, maternal strokes, myocardial infarction, abruptio placentae or antepartum fetal demise and also of properly managing fetal complications, such as FGR and acute fetal distress, by closely choosing the time and way of delivery [4,22]. Consequently, PubMed displays more than 10,000 research papers regarding screening strategies for preeclampsia. Some studies have focused on preeclampsia prediction using sFlt-1/PlGF ratio values in the first, second or third trimester [23,24,25,26,27,28,29,30,31,32], while other papers have concentrated on uterine Doppler scan as a predictor of HDPs [33,34,35,36,37,38]. Finally, we discovered two articles which focused on preeclampsia screening using a combination of sFlt-1/PlGF ratio with uterine Doppler ultrasound parameters [39,40].

Verlohren [23] was one of the first researchers to focus on sFlt-1/PlGF cutoffs as prognostic markers for preeclampsia, stating that ratio values of ≥88 predicted the early-onset form with 95% sensitivity, while ratio values of ≥110 predicted the late-onset form with 58.2% sensitivity. In this light, we obtained 100% sensitivity for a slightly lower cutoff (59.55). Cerdeira, in the Interventional Study Evaluating the Short-Term Prediction of Preeclampsia / Eclampsia In Pregnant Women With Suspected Preeclampsia (INSPIRE) study [24], provided valuable insights into the sFlt-1/PlGF ratio’s potential to predict preeclampsia and its significantly higher predictive accuracy compared to clinical judgement: sFlt-1/PlGF values of ≤38 classified the pregnancy as low-risk, while ratio values of >38 meant a high risk of developing preeclampsia in the next week, from 24 to 37 weeks of gestation. These cutoffs displayed 100% sensitivity, compared to 83% sensitivity by clinical practice alone. In our study, we obtained the same sensitivity for the sFlt-1/PlGF ratio. However, the sensitivity provided by clinical screening was only 26%. In the Prediction of Short-Term Outcome in Pregnant Women with Suspected Preeclampsia Study (PROGNOSIS), published 3 years earlier than the INSPIRE trial, sFlt-1/PlGF cutoff values of >38 from 24 to 37 weeks of gestation showed a slightly lower sensitivity for predicting preeclampsia within the next 4 weeks—66.2% [25].

A post hoc analysis of the INSPIRE study [26] established that ratio values of ≥85, determined between 24 and 36 weeks, presented a significant accuracy in ruling-in preeclampsia in the next 4 weeks after the assessment, with a positive predictive value of 71.4%. In this regard, the same study interval of 4 weeks after determination was also used in our research, providing the values of 59.55 as cutoff for preeclampsia and 102.74 for the severe form.

Concerning sFlt-1/PlGF values determined between 20 and 37 weeks and preeclampsia risk, Caillon [27] presented somewhat lower ratio values in the case of pregnancies later impaired by preeclampsia compared to our study: 69 ± 13 vs. a median value of 140.65. However, in Caillon’s study, the sFlt-1/PlGF ratio value was still significantly higher among patients with later preeclampsia compared to the control, for whom mean values were 32 ± 25, providing proof that sFlt-1/PlGF is a reliable marker of screening in a population of high-risk patients, where it can be used to accurately distinguish high-risk patients who need intensive monitoring from high-risk patients for whom hospitalization was not required, despite having a risk factor.

The PROGNOSIS Asia study [28] noted significantly increased sFlt-1/PlGF values among pregnant women who developed fetal adverse outcomes compared to control in the next 1 week (148.9 vs. 7.4), and 4 weeks (86.9 vs. 6.3) after ratio assessment, as well as in the case of pregnancies complicated by preterm delivery < 34 weeks and <37 weeks, in comparison to term delivery. Relatively similar sFlt-1/PlGF ratio values were presented in our study, where the cutoff for severe preeclampsia with the above-mentioned complications was 102.74. In a recent prospective research study [29], Jeon gave additional insight into the relationship between sFlt-1/PlGF values and the later development of fetal complications: high-risk pregnancies (with sFlt-1/PlGF values of >85) displayed significantly lower gestational ages at delivery, compared to low-risk pregnancies (with sFlt-1/PlGF < 38)—32 weeks vs. 35.79 weeks—and also a significantly increased prevalence of FGR—75.6%vs. 10.5%, *p* = 0.023—as well as a significantly longer period of hospitalization in the neonatal intensive care unit. The cutoff value of >85 as a predictor of preeclampsia severity and complications, such as preterm birth, was also used by Soundararajan [30] in the Real life outpatient biomarker use in management of hypertensive pregnancies in third trimester in a low resource SeTting (ROBUST) study, who found that pregnant women who displayed ratio values > the cutoff had a significant risk (*p* < 0.001) of developing severe preeclampsia (90.9% vs. 8% in the case of women with sFlt-1/PlGF values of <33) associated with preterm birth (32.6 weeks compared to 37.4 weeks, in the case of low-risk pregnancies). However, it is notable that the ROBUST study included only third trimester high-risk women, with a gestational age between 28 and 37 weeks.

Regarding sFlt-1/PlGF ratio’s ability to distinguish between established preeclampsia and uncomplicated gestational hypertension, Ciciu et al. [31] discovered, in a recent prospective research study, that mean sFlt-1/PlGF values were significantly higher in pregnancies already diagnosed with preeclampsia, compared to those already diagnosed with gestational hypertension (209.2 ± 138.77 vs. 46.08 ± 17.37, *p* < 0.001). However, this study focused on differences in sFlt-1/PlGF values between already-diagnosed cases of gestational hypertension or preeclampsia and not on the prediction of these HDPs. On the other hand, our study concentrated on the prediction of gestational hypertension at least 4 weeks before its onset. In this light, we reported significantly higher median sFlt-1/PLGF values in the preeclampsia group (140.65), compared to the gestational hypertension group (36.5). In a similar manner, Yang et al. [32] also used the sFlt-1/PlGF ratio to differentiate between gestational hypertension and established preeclampsia, obtaining significant differences. However, although gestational hypertension does not typically increase the risk of severe maternal or fetal morbidity, it carries an increased risk of evolving into preeclampsia, often in unpredictable circumstances. For this reason, we considered gestational hypertension screening as truly benefic, as patients discovered in the early second trimester to be carrying an increased risk of later gestational hypertension still need closer pregnancy monitoring. Consequently, we developed a gestational hypertension prediction algorithm with specific cutoffs.

As a whole, there are fewer published articles focusing on the role of second trimester ultrasound performance in HDP prediction compared to those whose central focus is on the role played by the sFlt-1/PlGF ratio. First of all, the vast majority of available papers from the literature have found mean uterine PI to be the main piece of preeclampsia prediction strategies. As previously mentioned in this paper, the 2018 ISUOG guidelines stated that the 95th centile of mean uterine PI of 1.44 at 23 weeks ultrasound scan had predictive value for preeclampsia [16]. This value is similar to 1.44, the cutoff we found in case of severe preeclampsia, and higher than 1.25, the cutoff established in our study for preeclampsia. Furthermore, Adekanmi [33] reported the mean uterine PI value of 1.34 in the case of pregnancies later affected by preeclampsia, which was significantly higher than the mean uterine PI of 0.75 in the control group. We reported higher mean uterine PI values in the cases of women who later developed preeclampsia and also higher control values (1.5 ± 0.13 vs. 1.12 ± 0.13), keeping the difference between them significant. Additionally, Trongpisutsak [34] also obtained increased mean uterine PI values at 16–24 weeks of gestation in the case of pregnancies later affected by preeclampsia (1.34 ± 0.52 vs. 0.98 ± 0.28 for control, *p* = 0.004)—the same mean value as the one reported by Adekanmi—and established the best cutoff value for mean uterine PI at 1.025. Abdel Razik [17] considered the 20–24 weeks Doppler scan as one of the finest predictors of preeclampsia, establishing 1.14 as the mean uterine artery PI cutoff for preeclampsia prediction, which is slightly lower than our cutoff (1.25).

Concerning ultrasound screening for HDP-related complications, Ratiu [35] found that bilateral high uterine PI and RI values at 19–22 weeks put the pregnancy at significant risk of developing preeclampsia, a small-for-gestational-age (SGA) birthweight < 10th centile and also FGR (birthweight < 3rd centile). We also highlighted the relationship between increased mean uterine PI and RI values at 24 weeks (with cutoff values of 1.44 and 0.69, respectively) and later-onset preeclampsia cases complicated with FGR. Barati [36] defined abnormal uterine Doppler at 16–22 weeks of gestation as a mean PI value of >1.45, describing significant correlations between abnormal uterine Doppler and the occurrence of early-onset preeclampsia (<32 weeks), with 95.5% specificity and 79% sensitivity (*p* < 0.001), as well as later development of SGA fetus (birthweight < 10th centile): 23.5% vs. 0.82% for the control and the preterm delivery groups (11.8%vs. 1.4%). For similar fetal complications, we reported a mean uterine PI value of 1.58 at 24 weeks.

Though considered less accurate in HDP screening than the mean uterine PI, the mean uterine RI also proved to be a relevant predictive marker. In this regard, Adekanmi [33] reported significantly higher mean uterine RI values at 20–24 weeks, in cases of pregnancies later imbalanced by preeclampsia (0.59 vs. 0.5 for control, *p* = 0.002). These values are lower than the mean uterine RI value of 0.68 reported in the present paper for preeclampsia prediction (0.68), while we found a similar mean RI value in the control group (0.5). In addition, Abdel Razik [17] also reported significantly higher mean uterine RI values in pregnancies later complicated by preeclampsia, obtaining a cutoff value of 0.61, very close to our cutoff value, which was 0.62. Maged [37] performed uterine artery Doppler ultrasound screening at 18–22 weeks and provided additional insight, finding significantly higher mean uterine RI values in cases of women who later developed preeclampsia (0.587 ± 0.072) or FGR (0.587 ± 0.053), compared to controls (*p* < 0.001). These RI values are slightly lower than those obtained in our study, where the mean RI value for preeclampsia was 0.68 ± 0.06. A very recent retrospective study conducted by Li [38] discovered that increased bilateral RI (Odds ratio (OR) 2.83) was an effective marker for preeclampsia prediction. Furthermore, every 0.1 increase in the median multiple of mean RI brought a 22% raise in the risk of preeclampsia.

Concerning the two papers that combined the sFlt-1/PlGF ratio with Doppler ultrasound for HDP screening, Diguisto [39] performed a multicentric study in which PlGF was reported to be a useful tool for predicting preeclampsia, while sFlt-1 or uterine artery Doppler indices did not bring any improvement. However, it is noteworthy that only first trimester pregnancies, with gestational ages from 11 to 13 weeks, were included in the study. On the contrary, in his retrospective paper, Graupner [40] included singleton pregnancies with confirmed late-onset preeclampsia (≥34 weeks), stating that the addition of sFlt-1/PlGF cutoff values of >110 to a mean uterine PI cutoff value of >95th centile improved the prediction of SGA babies. Our study had similar findings with the specific cutoffs we provided but was reported at another screening time (24 weeks).

Although uterine artery Doppler scans may improve the screening performance provided by sFlt-1/PlGF ratio at 24 weeks, mainly improving the prediction of severe preeclampsia with its specific complications, we consider it to be particularly useful in many healthcare systems affected by limited resources, where determining the sFlt-1/PlGF ratio values in all pregnancies considered to be high-risk can be unfeasible from a financial point of view, representing a financial burden, a situation also exposed in the study published by Oancea et al. [41].

Last but not least, The American College of Obstetricians and Gynecologists [4] highlighted the importance of first trimester preeclampsia prophylaxis with daily low-dose acetylsalicylic acid, starting ideally before 16 weeks, among women judged as being high-risk based on the traditional risk factors. However, pregnant women enrolled in the present study did not benefit from a standard, uniform first trimester screening of preeclampsia; therefore, aspirin was not indicated. The focus of the present paper was to develop an effective second trimester screening for HDPs, as a significant proportion of women are not routinely screened between 11 and 14 weeks of gestation. Thus, plenty of the studies previously mentioned in this paper focused on the effectiveness of early second trimester screening strategies.

### Strengths and Limitations

One of the main strengths of the present research is the development of second trimester prediction algorithms based on multiple parameters (the biochemical markers sFlt-1 and PlGF, as well as uterine Doppler ultrasound criteria). These algorithms may represent valuable screening tools used by physicians for better second trimester assessments of patients at a high risk of developing HDPs and HDP-related complications, compared to evaluations based solely on traditional risk factors, serum or Doppler parameters. In addition, an important advantage of the algorithms developed in the present paper is the fact that they offer prediction models for uncomplicated gestational hypertension and assess the severity of preeclampsia by providing personalized cutoff values for the mild form and the severe form, thus enabling an early effective prediction of severe and life-threatening obstetric complications. The foremost clinical application of these algorithms would be the effective second trimester prediction of early-onset severe preeclampsia with life-threatening obstetric complications. On the other hand, the results presented in this study may have been impacted by the relatively small number of patients enrolled and also by the single-center involvement. Further multicentric studies with larger samples would be needed in order to better quantify the improvement brought by the addition of mean uterine Doppler parameters to sFlt-1/PlGF ratio values for the second trimester prediction of HDP occurrence, severity and related maternal and fetal complications. Moreover, the use of a highly selected study sample (pregnant women displaying at least one traditional risk factor for HDP) limits the external validity of the study.

## 5. Conclusions

The sFlt-1/PlGF ratio and mean uterine artery Doppler ultrasound PI and RI values, determined at 24 weeks, significantly improve the second trimester prediction of preeclampsia (mainly the early-onset form), gestational hypertension and HDP-related complications among high-risk patients, compared to predictions based on traditional risk factors. The algorithm comprising the combination of a sFlt-1/PlGF cutoff of 102.74, with a mean uterine PI cutoff of 1.44 and a mean uterine RI cutoff of 0.69 enhances the prediction accuracy of severe, early-onset preeclampsia complicated by FGR and acute fetal distress with a CPR of <one, compared to the biochemical markers or uterine Doppler parameters used alone. At the same time, the algorithm based on the combination of a sFlt-1/PlGF cutoff of 59.55 with a mean uterine PI cutoff of 1.25 and a mean uterine RI cutoff of 0.62 significantly enhances the prediction accuracy for preeclampsia, compared to screening based on traditional risk factors. Finally, an algorithm based on the combination of a sFlt-1/PlGF cutoff of 59.55 with a mean uterine PI cutoff of 1.44 and a mean uterine RI cutoff of 0.74 significantly enhances the prediction accuracy of gestational hypertension. HDP screening strategies should increasingly focus on implementing such algorithms for women initially judged as high-risk based on their medical history and clinical factors, in order to properly diagnose HDPs and properly limit or manage later maternal and fetal complications.

## Figures and Tables

**Figure 1 children-11-00468-f001:**
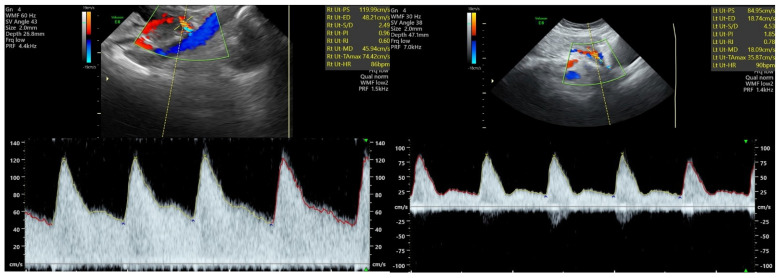
Second trimester Doppler ultrasound scan of the uterine artery at 23−24 weeks of gestation. (**Left**): normal pregnancy—PI and RI values within normal range. (**Right**): significantly increased values of PI and RI can be noticed in the case of a pregnancy later diagnosed with early-onset preeclampsia.

**Figure 2 children-11-00468-f002:**
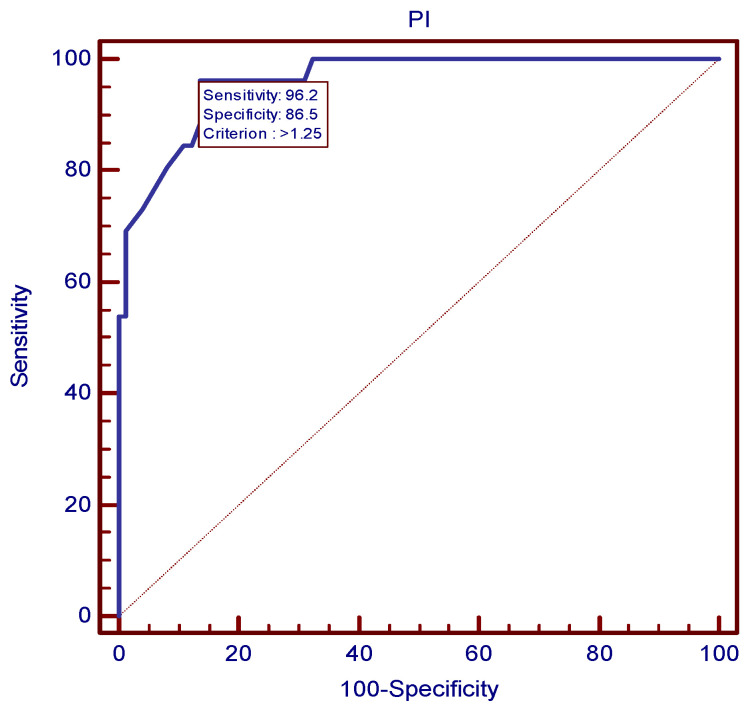
ROC curve for the most accurate uterine PI cutoff at 24 weeks for preeclampsia prediction.

**Figure 3 children-11-00468-f003:**
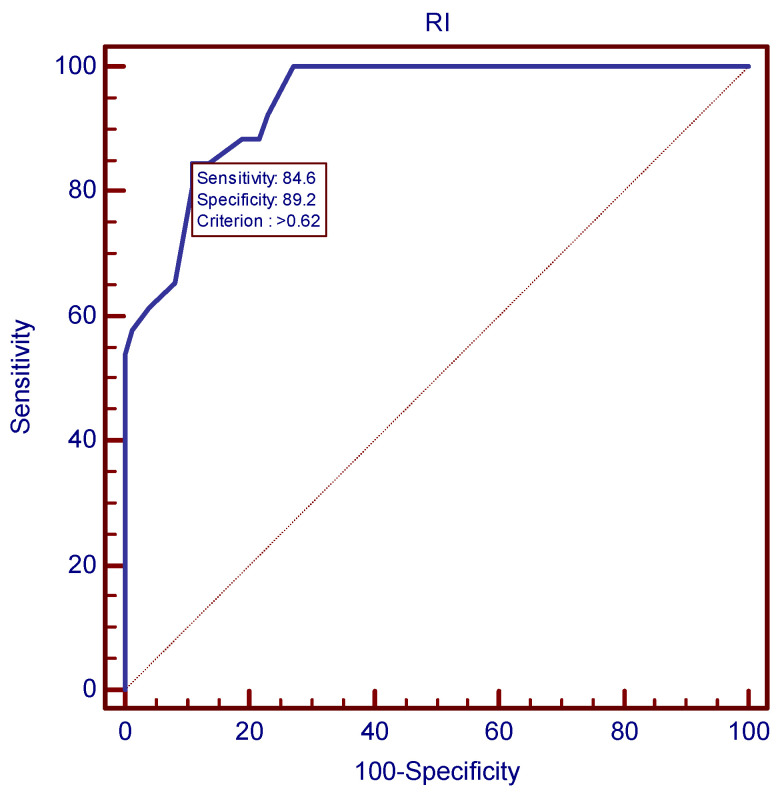
ROC curve for the most accurate uterine RI cutoff at 24 weeks for preeclampsia prediction.

**Figure 4 children-11-00468-f004:**
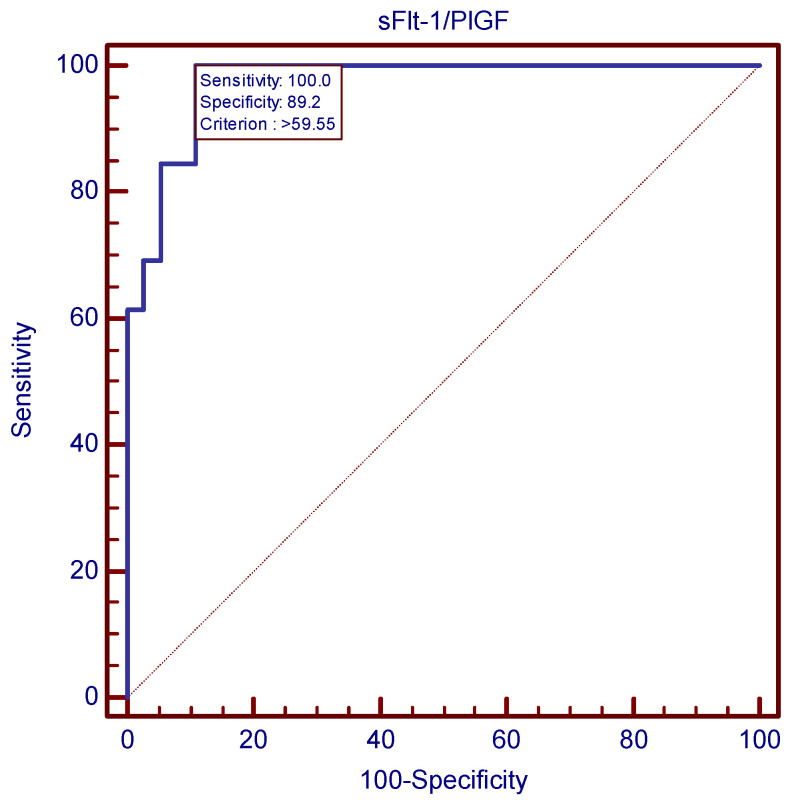
ROC curve for the most accurate sFlt-1/PlGF ratio value cutoff (59.55) at 24 weeks for preeclampsia prediction.

**Figure 5 children-11-00468-f005:**
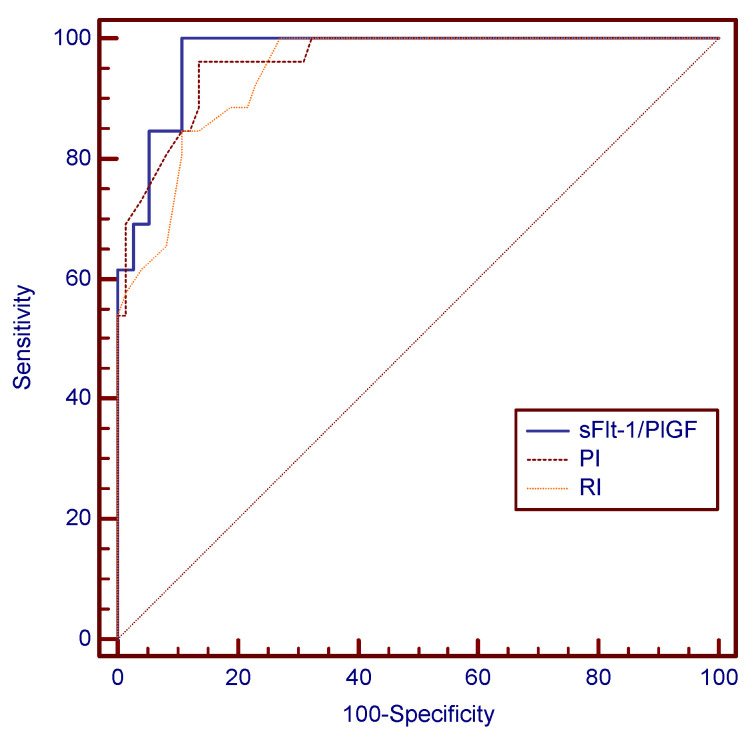
Comparison of the ROC curves for all three studied parameters—sFlt-1/PlGF is the most accurate parameter for preeclampsia prediction.

**Figure 6 children-11-00468-f006:**
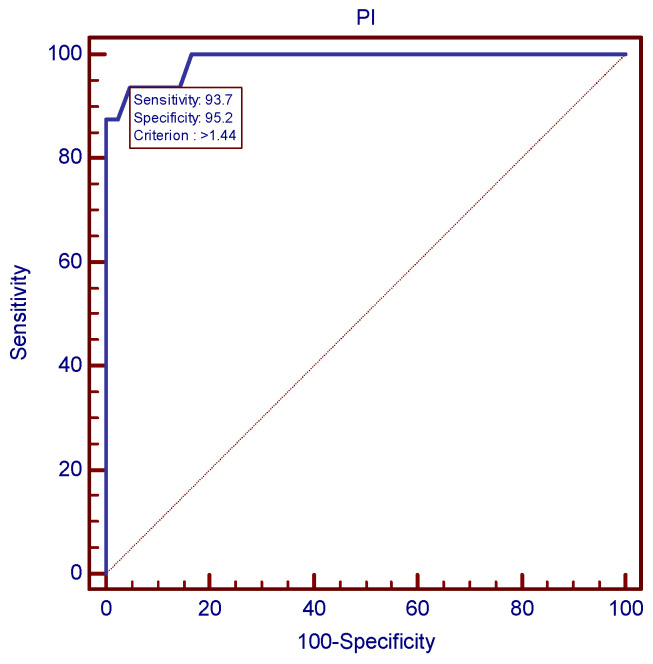
Severe preeclampsia prediction strategy: the ROC curve established 1.44 as the best uterine PI cutoff.

**Figure 7 children-11-00468-f007:**
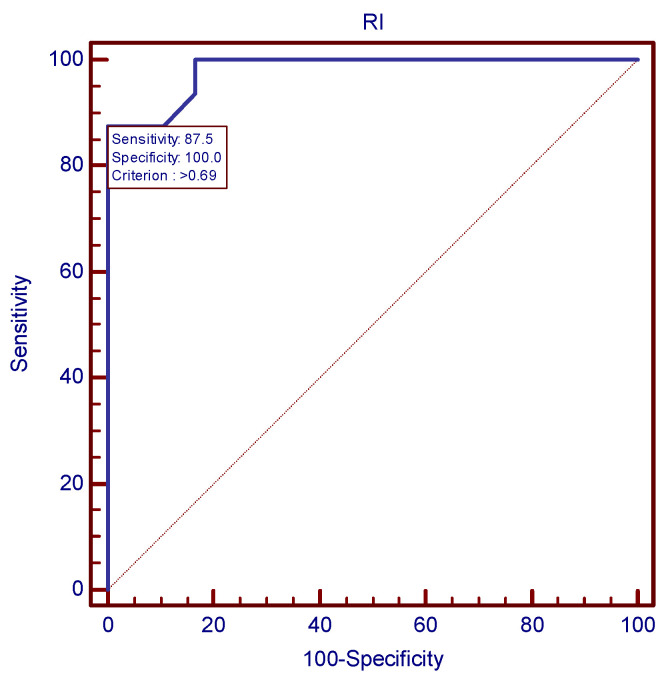
Severe preeclampsia prediction strategy: the ROC curve established 0.69 as the best uterine RI cutoff.

**Figure 8 children-11-00468-f008:**
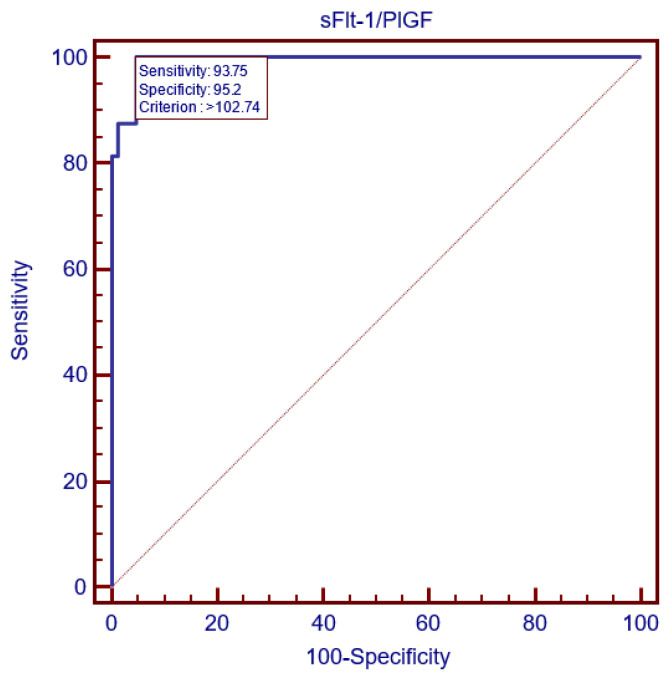
Severe preeclampsia prediction strategy: the ROC curve established 102.74 as the sFlt-1/PlGF ratio cutoff.

**Figure 9 children-11-00468-f009:**
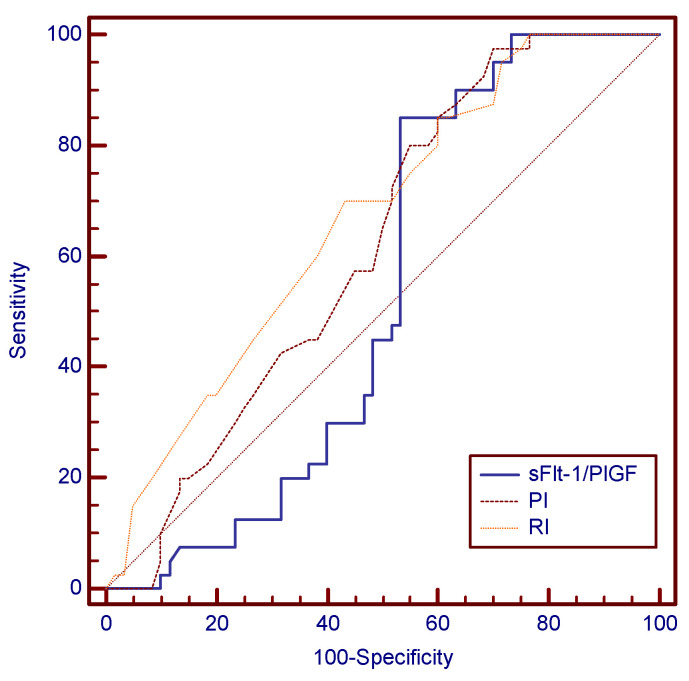
Graphical comparison of AUC for mean uterine PI, mean uterine RI and sFlt-1/PlGF—mean uterine RI is the best predictor of gestational hypertension.

**Table 1 children-11-00468-t001:** Prevalence of the three groups among the initial study sample: only 26% of the total number of patients categorized as carrying a highrisk for preeclampsia, based on clinical criteria, eventually developed the condition.

	Frequency (No.)	Prevalence (%)
Control group	34	34.00%
Gestational hypertension group	40	40.00%
Preeclampsia group	26	26.00%
Total study sample	100	100%

**Table 2 children-11-00468-t002:** Prevalence of the two forms of preeclampsia among the preeclampsia group: the vast majority developed the severe form.

	Frequency (No.)	Prevalence (%)
Mild preeclampsia group	10	38.46%
Severe preeclampsia group	16	61.54%

**Table 3 children-11-00468-t003:** Demographic data and clinical characteristics of the studied groups. Legend for Table 3: GA = gestational age, SD = standard deviation, w = weeks of pregnancy.

	Interval	Control Group	Gestational Hypertension Group	Mild Preeclampsia Group	Severe Preeclampsia Group
Age (mean ± SD)		26 ± 5.09	30 ± 5.55	31 ± 5.97	29 ± 6.29
Place of residence (no. of patients from the group)	Urban	27/34	29/40	7/10	11/16
	Rural	7/34	11/40	3/10	5/16
GA at onset (no. of patients from the group)	27–28 w				3/16
	29–30 w			2/10	
	31–32 w			7/10	13/16
	33–34 w			1/10	
GA at delivery (no. of patients from the group)	27–28 w				3/16
	33–34 w				13/16
	37–38 w	2/34	17/40	8/10	
	39–40 w	32/34	23/40	2/10	
Birthweight (g) (mean ± SD)	27–28 w				0.677 ± 0.129 (stillbirth)
	33–34 w				1382 ± 164
	37–38 w	3355.5 ± 325.97	3280 ± 412.58	3025 ± 181.19	
	39–40 w	3488.15 ± 231.46	3572 ± 381.23	3320 ± 218.16	
Way of delivery (no. of patients from the group)	Vaginal	28/34	28/40	2/10	3/16 (stillbirth)
	Caesarean	6/34	12/40	8/10	13/16

**Table 4 children-11-00468-t004:** Kruskal–Wallis test: median uterine PI values at 24 weeks were significantly different among the three study groups.

Mean Uterine PI Values at 24 Weeks	Control Group	Gestational Hypertension Group	Preeclampsia Group	*p*-Value
Number of values	34	40	26	<0.0001
Minimum	0.75	0.98	1.2	
Median	1.15	1.18	1.58	
Maximum	1.42	1.48	1.64	
Mean	1.12	1.19	1.50	
SD	0.13	0.13	0.13	

**Table 5 children-11-00468-t005:** Dunn’s multiple comparison test: median uterine PI values at 24 weeks were significantly higher among patients who later developed preeclampsia.

Dunn’s Multiple Comparison Test	*p*-Value
Control group vs. gestational hypertension group	0.0504
Control group vs. preeclampsia group	<0.0001
Gestational hypertension vs. preeclampsia group	<0.0001

**Table 6 children-11-00468-t006:** Kruskal–Wallis test: median uterine RI values at 24 weeks were significantly different among the three study groups.

Mean Uterine RI Values at 24 Weeks	Control Group	Gestational Hypertension Group	Preeclampsia Group	*p*-Value
Number of values	34	40	26	<0.0001
Minimum	0.42	0.42	0.54	
Median	0.52	0.52	0.70	
Maximum	0.68	0.69	0.75	
Mean	0.51	0.53	0.68	
SD	0.05	0.07	0.06	

**Table 7 children-11-00468-t007:** Dunn’s multiple comparison test: median uterine RI values at 24 weeks were significantly higher among patients who later developed preeclampsia.

Dunn’s Multiple Comparison Test	*p*-Value
Control group vs. gestational hypertension group	0.6393
Control group vs. preeclampsia group	<0.0001
Gestational hypertension vs. preeclampsia group	<0.0001

**Table 8 children-11-00468-t008:** Kruskal–Wallis test: median sFlt-1/PlGF values at 24 weeks were significantly different among the three study groups.

sFlt-1/PlGF Values	Control Group	Gestational Hypertension Group	Preeclampsia Group	*p*-Value
Number of values	34	40	26	<0.0001
Minimum	2.020	3.99	67.51	
Median	10.05	36.51	140.7	
Maximum	80.79	117.6	392.4	
Mean	15.34	40.56	157.9	
SD	17.57	30.92	85.43	

**Table 9 children-11-00468-t009:** Dunn’s multiple comparison test: patients who later developed preeclampsia displayed significantly higher median sFlt-1/PlGF values at 24 weeks.

Dunn’s Multiple Comparison Test	*p*-Value
Control group vs. gestational hypertension group	<0.0001
Control group vs. preeclampsia group	<0.0001
Gestational hypertension vs. preeclampsia group	<0.0001

**Table 10 children-11-00468-t010:** Comparison of the ROC curves for all three studied parameters. Legend for Table 10: AUC = area under the curve; SE = standard error; CI = confidence interval.

	AUC	SE	95% CI
sFlt_1_PlGF	0.973	0.0127	0.919 to 0.995
PI	0.961	0.0175	0.902 to 0.990
RI	0.944	0.0211	0.879 to 0.980

**Table 11 children-11-00468-t011:** Kruskal–Wallis test: median uterine PI values at 24 weeks were significantly different among the four study groups.

Mean Uterine PI Values at 24 Weeks	Control Group	Gestational Hypertension Group	Mild Preeclampsia Group	Severe Preeclampsia Group	*p*-Value
Number of values	34	40	10	16	<0.0001
Minimum	0.75	0.98	1.2	1.41	
Median	1.15	1.18	1.42	1.60	
Maximum	1.42	1.48	1.47	1.64	
Mean	1.12	1.19	1.37	1.58	
SD	0.13	0.13	0.09	0.06	

**Table 12 children-11-00468-t012:** Dunn’s multiple comparison test: patients who later developed severe preeclampsia displayed significantly higher median uterine PI values at 24 weeks.

Dunn’s Multiple Comparison Test	*p*-Value
Control group vs. gestational hypertension group	0.0504
Control group vs. mild preeclampsia group	<0.0001
Control group vs. severe preeclampsia group	<0.0001
Gestational hypertension group vs. mild preeclampsia group	0.0002
Gestational hypertension group vs. severe preeclampsia group	<0.0001
Mild preeclampsia group vs. severe preeclampsia group	0.0002

**Table 13 children-11-00468-t013:** Kruskal–Wallis test: median uterine RI values at 24 weeks were significantly different among the four study groups.

Mean Uterine PI Values at 24 Weeks	Control Group	Gestational Hypertension Group	Mild Preeclampsia Group	Severe Preeclampsia Group	*p*-Value
Number of values	34	40	10	16	<0.0001
Minimum	0.42	0.42	0.54	0.65	
Median	0.52	0.52	0.66	0.72	
Maximum	0.68	0.69	0.69	0.75	
Mean	0.51	0.53	0.62	0.71	
SD	0.05	0.07	0.06	0.02	

**Table 14 children-11-00468-t014:** Dunn’s multiple comparison test: patients who later developed severe preeclampsia displayed significantly higher median uterine RI values at 24 weeks.

Dunn’s Multiple Comparison Test	*p*-Value
Control group vs. gestational hypertension group	0.6393
Control group vs. mild preeclampsia group	<0.0001
Control group vs. severe preeclampsia group	<0.0001
Gestational hypertension group vs. mild preeclampsia group	0.0012
Gestational hypertension group vs. severe preeclampsia group	<0.0001
Mild preeclampsia group vs. severe preeclampsia group	0.0003

**Table 15 children-11-00468-t015:** Kruskal–Wallis test: median sFlt-1/PlGF values at 24 weeks were significantly different among the four study groups.

sFlt-1/PlGF at 24 Weeks	Control Group	Gestational Hypertension Group	Mild Preeclampsia Group	Severe Preeclampsia Group	*p*-Value
Number of values	34	40	10	16	<0.0001
Minimum	2.02	3.99	67.51	103.6	
Median	10.05	36.51	90.48	179.9	
Maximum	80.79	117.6	128.9	392.4	
Mean	15.34	40.56	89.69	200.5	
SD	17.57	30.92	22.75	82.56	

**Table 16 children-11-00468-t016:** Dunn’s multiple comparison test: patients who later developed severe preeclampsia displayed significantly higher median sFlt-1/PlGF values at 24 weeks.

Dunn’s Multiple Comparison Test	*p*-Value
Control group vs. gestational hypertension group	<0.0001
Control group vs. mild preeclampsia group	<0.0001
Control group vs. severe preeclampsia group	<0.0001
Gestational hypertension group vs. mild preeclampsia group	0.0001
Gestational hypertension group vs. severe preeclampsia group	<0.0001
Mild preeclampsia group vs. severe preeclampsia group	<0.0001

**Table 17 children-11-00468-t017:** Comparison of the ROC curves for all three studied parameters—mean uterine RI is the most accurate parameter for the prediction of gestational hypertension, displaying the highest AUC value. Legend for Table 17: AUC = area under the curve; SE = standard error; CI = confidence interval.

	AUC	SE	95% CI
sFlt_1_PlGF	0.529	0.0587	0.427 to 0.630
PI	0.616	0.0555	0.514 to 0.712
RI	0.666	0.0538	0.565 to 0.757

## Data Availability

The data presented in this study are available on request from the corresponding author. The data are not publicly available due to privacy of research participants.

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
