# Peer review of "A Second Trimester Prediction Algorithm for Early-Onset Hypertensive Disorders of Pregnancy Occurrence and Severity Based on Soluble fms-like Tyrosine Kinase 1 (sFlt-1)/Placental Growth Factor (PlGF) Ratio and Uterine Doppler Ultrasound in Women at Risk"

_children, 2024, doi:10.3390/children11040468_

Round 1

Reviewer 1 Report

Comments and Suggestions for Authors

It s a well prepared manuscript hovewer there are many controversial aspects within it:

1. First of all , if You studied a high risk group from the 24 weeks of gestation , why You did not mention antyhing about aspirin treatment ? Did they all receive aspirin or did they have all individual first trimester screening of preeclampsia (biochemistry + ultarsound + MAP) ? It s very important . You should mention about it in the introduction , discussion etc. The number of  patients with complications are quite high so You have to make comments about aspirin. What is your strategy in Romania? What kind of screening of PE is used. It is very important to write about it in the Introduction because generally the strategy You presented is not an universal , modern one which promotes different strategy - first trimester individual screening and eventual prophylaxis.

2. In the Methods You have to write down very precisly the inclusion criteria (all , step by step)

3. Do not use term IUGR if its only EFW below < 10th percentile . Try to calculate all percentages for FGR according to Delphi criteria

4. It lacks table with clinical characteristics!!! . It s crucial. What was GA at onset(diagnosis), GA at delivery, newborns weight , way od delivery, percentage of HELLP in each group!!!!

Author Response

All the revisions are highlighted in yellow.

Point 1: First of all , if You studied a high risk group from the 24 weeks of gestation , why You did not mention antyhing about aspirin treatment ? Did they all receive aspirin or did they have all individual first trimester screening of preeclampsia (biochemistry + ultarsound + MAP) ? It s very important . You should mention about it in the introduction , discussion etc. The number of  patients with complications are quite high so You have to make comments about aspirin. What is your strategy in Romania? What kind of screening of PE is used. It is very important to write about it in the Introduction because generally the strategy You presented is not an universal , modern one which promotes different strategy - first trimester individual screening and eventual prophylaxis.

Response 1: We added two paragraphs (one in Introduction and the other one in Discussion) regarding first trimester screening of preeclampsia and phrophylaxis with aspirin. In Romania, in selected cases patients considered as high-risk during screening at 11-14 weeks (based on the presence of traditional risk factors, blood pressure and uterine artery PI) receive daily low-dose of aspirin starting from 16 weeks. However, no patient included in this study received aspirin and they did not benefit from a standard first trimester screening. However, as mentioned in the article, in the majority of cases young primiparous patients, without evident risk factors, develop an early onset, severe form of preeclampsia, despite being regarded as carrying a low-risk at the initial first trimester screening – this situation applies in Romania, as well. Consequently, second trimester screening strategies are needed, too. This is why there are so many recent studies which start preeclampsia screening at 20-24 weeks of preganncy based on sFlt-1/PlGF ratio and Doppler ultrasound parameters. Therefore, we find the implementation of second trimester preeclampsia screening strategies to be very useful.

Point 2: In the Methods You have to write down very precisly the inclusion criteria (all, step by step).

Response 2: The inclusion and exclusion criteria are stated in the Materials and Methods section – Study population. Inclusion criteria: patients with singleton pregnancy, 24 weeks of gestation, with at least one of the traditional clinic or anamnestic risk factor for preeclampsia. We also added a few words about the absence of first trimester preeclampsia screening in the enrolled patients.

Point 3: Do not use term IUGR if its only EFW below < 10th percentile . Try to calculate all percentages for FGR according to Delphi criteria.

Response 3: Thank you very much for this very good observation. FGR definition was corrected. The 2020 ISUOG criteria for early FGR diagnosis were followed, as stated in the Materials and Methods section, in lines 166 – 170.     

Point 4: It lacks table with clinical characteristics!!! . It s crucial. What was GA at onset(diagnosis), GA at delivery, newborns weight , way od delivery, percentage of HELLP in each group!!!!

Response 4: A table comprising demographic and clinical characteristics was added – see Table 3 from Results. We also stated that no evidence of HELLP Syndrome or evolution to eclampsia was found among the study sample.

Reviewer 2 Report

Comments and Suggestions for Authors

The research paper entitled “A Second Trimester Prediction Algorithm for early-Onset Hypertensive Disorders of Pregnancy Occurrence and Severity Based on sFlt-1/PlGF Ratio and Uterine Doppler Ultrasound in Women at Risk” presents a second trimester screening cut-off for hypertensive disorders. Overall, the manuscript is well written.

My comments regarding the manuscript are:

-              My main concern is regarding the study sample. I believe that the sample is relatively small for a trichotomous comparison (control vs. gestation hypertension vs. preeclampsia) and due to that fact, the cut-offs set by the authors are in doubt. The authors should provide information on how they opted to this certain sample and the statistical power that this sample has.

-              Furthermore, the authors should provide demographic details for the groups in their study. In my opinion the selection of a matched control group would have been better.

-              The authors should explain the reason why preterm birth or CPR less than 1 solely are criteria for severe preeclampsia.

-              The authors should also provide information about the 16 cases of severe preeclampsia (etiology and outcome)

-              In lines 158 to 160 please correct the definition of fetal growth restriction. The definition provided by the authors refers to SGA fetuses. ISUOG defines FGR as a condition in which the fetus fails to reach its genetically predetermined growth potential and provides criteria for the definition.

Author Response

All the revisions are highlighted in yellow.

Point 1: My main concern is regarding the study sample. I believe that the sample is relatively small for a trichotomous comparison (control vs. gestation hypertension vs. preeclampsia) and due to that fact, the cut-offs set by the authors are in doubt. The authors should provide information on how they opted to this certain sample and the statistical power that this sample has.

Response 1: We mentioned in the Strengths and Limitations section about the relatively small number of patients enrolled and also the monocentric location of the study. We managed to include 100 patients regarded as high-risk for preeclampsia in the enrollment period. Some pregnant women could not be included in the study because they refused to give the informed consent. Moreover, especially in the first few months of study enrollment, the number of patients was negatively impacted by the COVID-19 pandemic and the challenges it brought in emplementing clinical studies. Furthermore, the study was monocentric, as there is another Obstetrics and Gynaecology Department in our university center, where patients with HDP are also treated and screened. However, the study design – including the trichotomous comparison - was approved by the statistician for the present study sample. The statistical results, including mean/median comparison and cutoff values as revealed in the Methods and Results sections, are valid.

Point 2: Furthermore, the authors should provide demographic details for the groups in their study. In my opinion the selection of a matched control group would have been better.

Response 2: A table comprising demographic and clinical characteristics of the studied groups was added – see Table 3 from Results.

Point 3: The authors should explain the reason why preterm birth or CPR less than 1 solely are criteria for severe preeclampsia.

Response 3: The criteria that patients had to match in order to be part of preeclampsia group and then considered as having the mild or severe forms of preeclampsia have been more clearly described in the Materials and Methods – Study Population section. Criteria for severe preeclampsia include maternal signs (blood pressure ≥160/110 mmHg, thrombocytopenia, newly developed headache or kidney failure) as recommended by 2020 ACOG practice bulletin number 222, as well as the presence of FGR leading to preterm birth, intrapartum fetal demise or acute fetal distress with CPR <1.     

Point 4: The authors should also provide information about the 16 cases of severe preeclampsia (etiology and outcome).

Response 4: Information about the outcome was provided in the Results section - all pregnancies with severe preeclampsia were affected by FGR with signs of acute fetal distress that led to premature delivery and in 3 out of 16 cases, intrapartum fetal demise. Further information regarding the way of delivery was added: All severe preeclampsia cases delivered through caesarean section, except the three pregnancies complicated by intrapartum fetal demise, in whom labour was induced with systemic Oxytocin. This data was represented in Table 3. Regarding the etiology, as added in the Discussion section, in many cases young primiparous patients, without evident risk factors, developed an early onset, severe form of preeclampsia, despite being regarded as carrying a low-risk for the condition. The etiology of severe preeclampsia in the 16 cases – what triggers the narrowing of the maternal spiral arteries and subsequent placental hypoperfusion -  remains largely unknown. However, the 16 patients displaying the severe form of preeclampsia carried at least one of the traditional risk factors, which are mentioned in the Introduction and Methods.    

Point 5:  In lines 158 to 160 please correct the definition of fetal growth restriction. The definition provided by the authors refers to SGA fetuses. ISUOG defines FGR as a condition in which the fetus fails to reach its genetically predetermined growth potential and provides criteria for the definition.

Response 5: Thank you very much for this very good observation. FGR definition was corrected. The 2020 ISUOG criteria for early FGR diagnosis were followed, as stated in the Materials and Methods section, in lines 166 – 170.

Round 2

Reviewer 1 Report

Comments and Suggestions for Authors

You did include the recommended changes what makes the manuscript of higher value. Hovewer I do recommend some small changes:

_  in the Introduction You have to mentioned also about biochemical markers  which are part of PE screening (from line 127)

- I would not write in the discussion that many young healthy patient develop PE instead of low risk in the screening, because its not true . Prediction of PE < 34 weeks by history , MAP< uterinbe arteries doppler and PLGF is 90% ! So I do not agree with it . I would delete it.  

Author Response

Point 1:  In the Introduction You have to mentioned also about biochemical markers  which are part of PE screening (from line 127).

Response 1: We mentioned the biochemical markers – PlGF and PAPP-A which are part of first trimester preeclampsia screening.

Point 2: I would not write in the discussion that many young healthy patient develop PE instead of low risk in the screening, because its not true . Prediction of PE < 34 weeks by history , MAP< uterinbe arteries doppler and PLGF is 90% ! So I do not agree with it . I would delete it.  

Response 2: We have deleted this statement.

Reviewer 2 Report

Comments and Suggestions for Authors

The authors sufficiently addressed the issues raised.

Author Response

Thank you very much for all the advice.